# Elastic Rituals: A Multi-Religious Analysis of Adaptations to the COVID-19 Crisis

**Monica Cornejo-Valle * and Borja Martin-Andino**

Department of Social Anthropology and Social Psychology, Complutense University of Madrid,
28223 Madrid, Spain; borjamartinandino@ucm.es
* Correspondence: mcornejo@ucm.es

**Abstract:** The COVID-19 crisis truly challenged social interaction, the use of space and objects, as well as our sense of purpose and meaning in life. In this context, religious communities faced sudden interruption of their usual activities, lack of access to communal spaces and a global epidemic that summoned ancient "medieval plague" anxieties to work with. This article focuses on the vast repertoire of adaptations and reactions to the crisis that several religious communities developed in Spain. Our research is based on 40 conversations with members of Protestant and Evangelical denominations, Sunni Islam, Orthodox churches, Jehovah's Witnesses, Church of Jesus Christ of Latter-day Saints, Judaism, Hinduism, Buddhism, Sikhism, Church of Scientology, Baha'i, Seventh-day Adventist Church, Christian Science and Paganism, all of them minorities in the traditionally Catholic country. To analyze this repertoire of adaptations we focus on three aspects: the general context of changes and challenges, the ritual adaptations and the subjective experience of the adaptations. Grace Q. Zhang's theories on linguistic elasticity will be applied to understand the elasticity of ritual adaptations in COVID times.

**Keywords:** religious change; pandemic; ritual theory; COVID-19; Spain; religious minorities; multireligious comparative

## 1. Introduction

As Ronald L. Grimes has recently pointed out (Grimes 2021), even today the dominant conception of ritual in Anthropology, and in the Sciences of Religions in general, evokes rigidity as one of its characteristic attributes. According to a significant number of definitions, rigidity, as opposed to flexible or elastic, seems to be still an element that is academically used to define particular behaviors as ritual behaviors, which might lead us to presume that a title such as "elastic rituals" contains a substantial antithesis. However, in the last decades, we can find some literature that is especially focused on revealing ritual flexibility, by paying attention to those forms of religiosity in which the participants are expected to be ritual creators, as occurs in some contemporary paganisms and New Age spiritualities (Magliocco 2001; Trulsson 2010; Fedele 2012), as well as in Christianities (Csordas 1997; Roussou 2013). Although this literature started in the 1990s of the 20th century, flexibility in rites is not a characteristic of contemporary religions in the West. This new approach has just found that negotiations, invention and creativity in ritual practices may be traced all across different religious traditions and different times (Palmisano and Pannofino 2017; Hüsken and Neubert 2011). This article aims to contribute to this literature through the analysis of ritual adaptations in pandemic times, applying Grace Q. Zhang's theories on linguistic elasticity to explore how rituals are flexible and negotiated.

In 2020, religious practices and beliefs were quaked by the coronavirus crisis and the regulations regarding social life during the pandemic period. The use of places of worship, the distribution of information on the streets, family rituals and all kinds of community celebrations underwent changes, cancellations and strategic adaptations, with different

results depending on the needs, resources and preferences of each religious community. The period gave us a privileged observation of massive religious improvisation, when every community was pushed from the outside into changing something. The recent literature on COVID-19 and religion has shown some interest in ritual adaptations (Roso et al. 2020; Higgins and Djupe 2022); although, it is more generally devoted to the discussion about the role of religiosity in the population response to health policies, social distancing, prevention measures and vaccination.

Both in Social Sciences and Health–Medicine literature, the role of religion in the pandemic seems to be generally approached from two major themes that we can summarize as the theodiceal and the secular, or, as New York Times journalist Vivian Yee nicely put it: "In a Pandemic, Religion Can Be a Balm and a Risk" (Yee 2020). The theodiceal theme considers the idea of religions and spirituality helping people to deal with the anxiety of the COVID crisis, studying how religion might be (or not) a repertoire of coping mechanisms for resilience, mental and physical health (Levin 2020; Usarski and Py 2020; Huygens 2021; Giménez-Béliveau 2021; Jaysawal and Saha 2022), while the secular theme explores the idea of religions as a source of more or less irrational behavior that might lead (or not) people against health recommendations (Upenieks et al. 2021; Baker et al. 2020; Griera et al. 2022). Often, some authors consider both perspectives at the same time (Schnabel and Schieman 2022) and both offer interesting insights into how religious beliefs, practices and institutions adapt themselves (or not) to the secular and post-secular context of contemporary societies. However, they do not generally pay attention to the adaptability itself, with the exception of some analyses about the virtualization of religious practices during the lockdowns (Gutiérrez and de la Torre 2020; Parish 2020; Bryson et al. 2020; Kühle and Larsen 2021; Ganiel 2021; Sabaté-Gauxachs et al. 2021). This specific literature explores the mediatization of religious resources, the digitalization of churches and religious communities, and other topics already developed in the pre-COVID-19 approaches to religion "and the internet", as Heidi A. Campbell put it (Campbell 2013), and some also offer interesting insights about the relevance of community engagement (Parish 2020; Bryson et al. 2020; Kühle and Larsen 2021; Frei-Landau 2020; Lee et al. 2022). Our point of view, however, is that the role of religiosity cannot be reduced to the theodiceal and secular perspectives, but is negotiated inside the communities.

While the pandemic context gave us a unique opportunity to compare how different communities, with different traditions and customs, respond to similar constraints in a determined context and period of history, the multi-religious perspective remains underdeveloped in the current literature. A systematic search on the Web of Science database shows that the vast majority of the studies published between 2020 and mid-2023 are focused on one denomination only, while studies deliberately oriented to compare more than one denomination are limited to Ho and Li (2022), which give us an effective idea of how little attention this perspective has received. Likewise, we miss a general theoretical framework with some perspective of universal applicability (if not validity) that eventually allows us to share a collective discussion on how rituals (and religions) adapt and update their means and practices from a broader comparative point of view. In order to go deeper into both dimensions, descriptive and theoretical, we would like to extend our study on the impact of the COVID crisis on religious minorities in Spain (Cornejo-Valle et al. 2022) and put our attention on the ritual adaptations and the theory that might help to understand them.

In 2022, we conducted 40 interviews with members of more than 20 different denominations in Spain, including Protestant and Evangelical denominations (Episcopalian, Pentecostal, Baptist, and Brethren Assemblies), Sunni Islam, Orthodox churches (Russian, Ukrainian from the Kyiv's Patriarchate, and Romanian), Jehovah's Witnesses, Church of Jesus Christ of Latter-day Saints, Orthodox and Reform Judaism, Vaishnava Hinduism, Buddhism (Soka Gakkai, Vajrayana), Sikhism, Church of Scientology, Baha'i, Seventh-day Adventist Church, Christian Science, Paganism (Wicca, Goddess worship) and Candomblé, all of them minorities in a traditionally Catholic country. The interviews were conducted both online and in person and were aimed at discovering the specific impact of the pan-

demic situation on religious minorities as part of the study titled, 'The Impact of the COVID-19 Crisis on Religious Minorities in Spain: Challenges for a Future Scenario' (COV-MINREL/2021, Foundation for Religious Pluralism and Coexistence). The sample excluded the Catholic community, which differs significantly from the other groups due to their cultural hegemony, social presence and material resources in Spain. The selected groups were chosen based on their demographic and legal presence in the Register of Religious Entities of the Ministry of Justice (available at http://www.mpr.gob.es, accessed on 30 April 2023).

While conducting the research, the interviews revealed a fascinating and complex variety of ritual adaptations that deserve specific attention. Considering the diverse sample, different communities responded with various strategies based on their own priorities: some emphasized the quality of the ritual according to tradition, while others focused on maintaining community continuity or even increasing their numbers if possible. Some communities tailored their adaptations based on an experimental sense of satisfaction or dissatisfaction that emerged during the adjusted celebrations. Throughout the research, we discovered that these types of responses strongly align with recent theories about elasticity in linguistics.

The concept of elasticity in Linguistics is a new perspective on vagueness in communication, especially developed by Grace Q. Zhang (2011, 2015). The value of Zhang's approach consists of adopting a pragmatic perspective on how vague language is negotiated in context. According to Zhang, the meanings of vague terms in speech acts are elastically stretched in order to satisfy the speakers' needs of communication, in an attempt to suit emerging new circumstances or conditions through negotiating meanings (Zhang 2011, p. 578). This implies that both the communication and the stretching of the terms are conditioned by the socio-cultural environment, because the speaker's needs, the emerging circumstances and the negotiation are usually defined according to cultural and social considerations for a particular context. However, elasticity is considered a universal property of languages (as vagueness is), and Zhang argues that its characteristics and functioning might be universal too.

Zhang's theory delves into the nuances of linguistic elasticity, such as the stretchers, the corresponding rules or maxims that they adhere to, and how they satisfy pragmatic functions. However, applying a theoretical framework to ritual behavior needs to start acknowledging that rituals are not the same as linguistic communication, even if many anthropologists have traditionally viewed rituals through the lens of semiotics, semantics, pragmatics, etc. According to the data we gathered, rituals are not communicative per se, as we can see in practice such as meditation, for example, whose implementation does not involve any kind of message exchange between the meditator and the sacred or the community, or another human or non-human agent. Likewise, since rituals are complex systems of practice and relationships, many aspects of rituals are performed with some sense of sensory esthetics that does not involve message exchange necessarily (smells, sounds, silence, light, darkness, etc.). At the same time that these non-communicative aspects can combine well with other ritual parts that might be communicative in essence (such as prayers to gods, exhortations, naming, etc.). Considering that rituals are not necessarily communication acts nor linguistics in essence, we will adapt Grace Zhang's theory to the particular nature of rituals, reformulating the function of elasticity for our case and reconsidering the stretchers and maxims that could help us to understand ritual behavior. Before that, we will describe the empirical context and main findings of our comparative study.

## 2. Navigating the Religious Lockdown in Spain

During the global crisis of 2020, each country implemented its own responses and, in particular, placed different emphases on the strictness of home confinement. Spain was one of the countries where the lockdown was stricter and lasted longer. From 14 March 2020, until the same month in 2022, mobility restrictions were not fully lifted in all parts of the

country. Religious communities experienced this situation with the same uncertainty and concern as the rest of the population. There was never a law that required the closure of places of worship, but the vast majority of them did close their doors in order to facilitate the containment of the virus through in-person contact. Some religious communities responded quickly by focusing on their charitable tasks and replacing meetings and rituals with other activities. Other communities moved their ritual and communal activities to virtual spaces, replicating their practices online as much as possible. Some trusted that the situation would be temporary and that the crisis would not last more than a few weeks, so they did not make any adaptations at first. However, over time, all communities had to incorporate the mandatory health measures that were imposed in the country: restrictions on public space mobility (lockdown, curfew), social distancing, capacity limits in shared spaces, social gathering restrictions, hydro-alcoholic gels and masks, proof of vaccination, ban of waters and baths, ban of shoe racks in temples, and even prohibition of singing in some cases (Cornejo-Valle et al. 2022).

All these measures challenged the traditions and customs of the majority of the religious communities in the country, as many of them depend on their worship places to gather the community and celebrate their rituals. Considering the scope of our sample, we found that the religious practices that were seriously affected by the anti-COVID measures can be summarized in eight major categories:

1. Gatherings at temples or worship places, which include a wide range of practices such as masses, worship, collective prayers or studying, in addition to community and family rituals such as funerals, weddings, baptisms and other rites de passage that might be celebrated in these shared spaces.

2. Activities and celebrations in open public spaces such as streets, but also natural places (woods, rivers), including practices such as distribution of information, public praying, processions, dancing, offerings or goddess worshiping, among other events.

3. Regular domestic celebrations and gatherings that some communities weekly or biweekly celebrate at their members' homes (such as Baha'i and Soka Gakkai or in other small communities).

4. Counselling and in-person services by pastors and community leaders or supporters, which is very popular among Evangelicals and Adventists, but also among some Buddhist, Christian Science, Scientology and Afro-American religions such as Candomblé.

5. Tithe, zakat, charity giving and other economic contributions that used to be attached to the service attendance in worship places. For some communities, the economic contribution might be a fundamental part of the prescriptions and rituals, as it is expected to be among Muslims and some Christian denominations.

6. Cleaning, baths, purification, baptism and other practices with water are also key for some ritual traditions, especially in Islam, Judaism, Sikhism and some Christian denominations such as Jehovah's Witnesses, but they are also important in some new traditions such as Scientology. These practices are usually performed in special places, well equipped for it, but the anti-COVID-19 measures included a ban on access and use of these places.

7. Communion in different aspects, such as food sharing in traditional celebrations (Ramadan, Christmas, Sucot, Vesak, and Diwali), or chanting, which might be a key aspect of the rituals in many Christian denominations as well as in some Hindu traditions (such as Gaudiya Vaishnavism). Orthodox, Anglicans and other Christians also celebrate rituals of communion with a sacramental sense of the sharing.

8. Farewells rites, wakes and funerals gained a special and poignant importance during the pandemic. While other gatherings could be canceled or rescheduled, cremations and burials had to be carried out despite restrictions such as capacity limits, bans on water use and the mandatory cremation of all bodies that were considered infected with COVID-19. These limitations created challenging situations where families could not perform traditional rituals such as a proper wake, or the washing or embalming

of the deceased. Additionally, for several months, the number of attendees allowed at funerals was limited to three people.

*2.1. Multi-Religious Ritual Adaptations*

The responses to these challenges varied widely depending on the doctrinal and traditional preferences of each religious community. The first significant adaptations are related to the limitations imposed over the gatherings at temples and worship places, most of which were closed by the communities to prevent the spread of the virus since the first weeks. Only Orthodox churches and some mosques remained open, and their main services (mass and communal prayer) were conducted. However, community events around family rituals, such as funerals, weddings, child welcoming ceremonies and other *rites de passage,* were canceled and rescheduled where possible.

To address the challenge, the majority of the religious organizations in Spain turned to technology in order to facilitate virtual gatherings of the local community. These adaptations include online celebrations and speeches by imams, priests and other community leaders through streaming and podcasting. In many cases, the local community engaged in international online conferences with other members of their religious network. Some communities embraced the virtualization enthusiastically, such as Christian Science, but others were reluctant to replicate the worship by video call, such as Sunni Muslims, who were recommended to engage in private prayer at home. When the mobility restrictions were lifted and some people started timidly going back to the worship places, the adaptations adopted were more in the fashion of the prophylactic measures that were already mandatory for public spaces across the country (from libraries to shopping centers). Those measures included social distancing, mask-wearing, space limitation, hand sanitizer, regular cleaning of the space and objects, hand washing and proper ventilation. When the worship places had some outdoor space, public or private, some communities took advantage of it to put some chairs outdoors and enlarge the capacity of the gatherings without contravening the measures.

The restrictions of mobility on public spaces affected especially those communities that include some sort of apostleship as part of their activities, such as Jehovah's Witnesses, Latter-Day Saints and other Christian groups. In these cases, the organizations recommended turning to the evangelization effort from the streets to the few spaces where interaction was ongoing, such as workplaces and social networks. Moreover, Pagans and the practitioners of Afro-American religions were affected by these restrictions since they celebrate particular rituals in open spaces such as woods, rivers, mountains, or just open lands with some soil to contact with the ancestors or Mother Earth. For some time, these practices were not possible unless the person owned land him or herself. Since that was not usually the case, the collective activities were canceled or postponed, but it is usually admitted in those traditions to have small altars at home and these were widely used and shared on social networks during the pandemic. When the restrictions to mobility eased, some of these communities came back to open public spaces such as parks, while others, such as Witnesses and Latter-day Saints did not come back to street preaching immediately. They preferred to wait until 2022, when the health situation was safer.

Some other groups reacted to the restrictions on mobility with an increase in their charity work in the cities. This charity work was allowed by the authorities when it was linked to the fight against the pandemic, and many small Christian churches, Islamic associations, the Scientology Foundation, and other religious communities came together to provide support to those in need as well as the authorities. They distributed masks, delivered food to elderly people who lived alone or could not leave their homes, donated medical supplies, and other forms of assistance to the community.

A couple of practices moved to virtual spaces easily as soon as the strict restrictions of mobility started: the regular domestic gatherings and the in-person counseling sessions. Community leaders adapted to the new situation by using conventional telephone and video calls to continue offering their one-to-one services to their fellow members. For those

who regularly celebrate religious meetings at home, online gatherings became the most common adaptation. In our inquiry, we found that two groups (the Soka Gakkai and Baha'i communities) were particularly well suited to this type of practice and easily adapted their traditions to group video calls. However, some ritual elements such as chanting together or sharing food were missed. Eventually, as the laws eased and good weather permitted, groups of people were able to meet in open spaces such as parks or outdoor areas of bars and restaurants, and these practices were also adopted in this format whenever possible.

The pandemic also had a significant impact on the economic aspects of communal life, according to our interviewees. In 2020, many religious organizations relied on attendance at services to generate cash contributions from their communities. However, with the exception of the Church of Latter-day Saints, which had already transitioned to electronic transfers in previous years, the pandemic prompted the adoption of online banks and money apps as new tools for collecting tithes, zakat, and other contributions for community and charity. Nonetheless, some people struggled to manage the new apps and digital tools for financial transactions. In these cases, community leaders met discreetly with members in person, often on the street or in the supermarket, to collect cash contributions. Since some communities consider economic exchanges linked to personal relationships and service attendance, the separation of these elements through digitalization brought a sort of de-ritualization of this aspect of community life. Despite these efforts, representatives from all 20 communities we interviewed reported a significant decrease in income during the pandemic. Other celebrations and gatherings related to economic contributions, such as fundraising events, were canceled or postponed until it was safe to gather again.

As previously mentioned, practices that involve cleaning, baths, purification, baptism, and other uses of shared water, were also affected by the anti-COVID measures in Spain. Water is an important ritual element in many religions. Mosques, synagogues and gurudwaras usually have a space for ablutions and ritual washing. Scientology employs saunas and baths for purification, while Witnesses and other Christians use large pools for baptisms, as well as smaller baptismal fonts. In 2020, all of these practices were prohibited. Imams recommended performing wudu at home before going to the mosque; although, Sikhs continued washing their hands in the gurudwara, understanding that they were also following anti-COVID measures, just in a ritual fashion. Although Orthodox churches remained open, baptisms were canceled as a general rule. When mobility restrictions were lifted, the ban on shared water remained in place for public places such as worship centers and hotels, but not for private spaces such as family homes. Those Christians that use to be baptized in pools managed to resume baptisms within the community with the help of those members who had swimming pools at home.

The closure of the worship centers and measures such as social distancing or mobility restrictions critically impacted all practices that require personal contact and co-presence. These practices, broadly labeled as communion, were attempted to be replicated through online meetings, but according to the leaders we interviewed, the experiences were generally disappointing. Sharing food through screens during Ramadan or Christmas did not feel like true communion. Hindu chanting and Christian choruses were constantly spoiled by the time lags in the calls, and were often canceled after several unsuccessful attempts. Those who have a sacramental sense of communion did not even try to replicate something that was not possible to recreate without the presence of the body and soul of the people. However, many still broadcasted masses and other celebrations via streaming and other means.

Finally, funerals and related practices and celebrations were also significantly impacted, and available adaptations were limited due to laws and norms related to public health. Given the constraints, and in line with the general tendency to make everything virtual, some communities opted for virtual gatherings to honor and mourn their deceased members, using the same format as their usual in-person services. Community leaders offered comforting services by telephone as well as online too. At burial sites and crematories, only family members or close individuals were allowed to attend. Some communities

preferred burial over cremation, and the Jewish community in Spain actively negotiated with municipalities to avoid cremation. We do not have notice of any adaptation related to the ritual washing and embalmment, which were not allowed and were, as a consequence, just canceled.

Summarizing the description above, we could synthesize the impact and adaptations as Table 1 shows:

**Table 1.** Ritual adaptations.

| Categories | Practices | Adaptations |
|---|---|---|
| Gatherings at worship places | Masses, worship, collective prayers or studying, community and family rituals such as funerals, weddings, baptisms and other *rites de passage* | Closure of worship places<br>Private prayer<br>Online replication<br>Streaming and podcasting<br>Charity work<br>Cancellation and reschedule<br>Prophylactic measures<br>(Eventually, outdoor spaces) |
| Activities in open public spaces | Distribution of information, public praying, processions, dancing, offerings, gods and ancestors worshiping in nature, other events | Cancellation<br>Evangelization in social media<br>Preaching at work<br>Online gatherings<br>Domestic altars<br>(Eventually, streets preaching and park altars) |
| Regular domestic celebrations | Prayer, study, food sharing | Online meetings<br>(Eventually, outdoor meetings) |
| Religious counseling | One-to-one and in-person support | Telephone and video calls<br>(Eventually, open spaces meetings) |
| Fundraising | Contribution at service<br>Charity giving<br>Funding events | Transition to electronic operations<br>Occasional meetings in person<br>De-ritualization |
| Water use | Cleaning, baths, purification, baptism, sauna, ablutions | At-home washing and purification<br>At home baptism |
| Communion | Food sharing in traditional celebrations<br>Chanting and choruses<br>Sacramental communion | Failed attempts of online replication<br>Cancellation<br>Prophylactic measures |
| Funerals | Farewells rites, wakes and funerals<br>Cremations and burials<br>Washing or embalming | 3 Attendees' limitation<br>Online honoring and mourning<br>Consent of cremation<br>Cancellation of body preparations and wakes<br>Prophylactic measures |

As can be seen from the above, our use of the term "ritual" in considering the effects and adaptations of measures against COVID-19 is rather loose. The breadth and diversity of our sample make it very difficult to apply one particular native concept of ritual, since different traditions conceive their liturgies (their nature, their limits, and their execution) in very different ways, which can only be unified from an external point of view. In this sense, if we had to make explicit an operative concept of ritual for this comparison, we would say that by ritual, we refer to culturally formalized and meaningful behavioral devices oriented to the organization of human experience of the sacred in the world. As our description of adaptations shows, this includes many different behaviors. What they all have in common is a significant degree of formalization that is recognized by each community as a meaningful way of relating to the sacred, whether they constitute easily

recognizable ritual units (masses, weddings, etc.) or just parts of larger ritual ensembles (washing, communion, etc.). Some of the practices outlined in our description may not be qualified as 'ritual' in some traditions (such as study, counseling or fundraising events), but sometimes they include recognizable ritual components (i.e., blessings, prayers, offerings or summons), and they may also show a more formalized display in other traditions (study is ritualized in Christian Science and some Buddhists communities, for example). We chose to include all these types of practices in order to gain a better perspective on the adaptations, with the conviction that a strict delimitation of ritual behavior is not coherent with research on elasticity as a quality of these very behaviors.

While from the point of view of the impact of measures against COVID-19, we can see eight different aspects of religious practices that were affected by the pandemic; from the point of view of adaptations, we observe only five major strategies, including the cancelation of activities as a non-adaptative response or negative adaptation:

- The cancellation and rescheduling of all those activities that could not be adapted (according to the criteria of each community), including the cancellation of practices such as the use of water and others imposed by the state's measures.
- The "domification" or confinement to the domestic spaces, and the use of private and family spaces for community activities that were previously developed elsewhere (places of worship mostly).
- The digitalization and virtualization of community life and its interactions, which, in a way, has meant the generalized occupation of a new space for interaction (virtual space) that had only been partially occupied before, but also involves the incorporation of digital media for religious practices that were previously conducted physically.
- The adoption of the generally recommended prophylaxis measures, with special importance given to capacity and social distancing in situations such as funerals and other encounters that were allowed over time.
- The outdooring and resignification of public and open spaces as places of religious significance, especially when anti-COVID measures eased. This includes a special use of streets for some charitable activities in the strictest lockdown times, but also a more general use of outdoor areas and parks as places of encounter and celebration for the communities.

### 2.2. The Subjective Experiences of the Adaptations

In order to explore the elasticity of ritual practices, it is relevant to consider how the people and their communities experienced the reactions to the pandemic challenge in the form of adaptations. These subjective experiences provide some sense of the satisfaction and dissatisfaction (resonating with John Austin's theory of speech acts, Austin 1975) with different adaptations and different aspects of them, as will be clear in the *verbatims*. Hence, in this section, we consider the subjective reception of the five major strategies of adaptation according to the communities' points of view, and through the voices of some who generously shared their reflections with us.

Cancelations, rescheduling and just the refusal to adapt the conventional rituals due to the anti-COVID measures seemed to have been very well received by many people in the first weeks of confinement, when they felt the cancelation as a stress reliever in their busy lives. However, in time, a widespread nostalgia for community life and a negative perception of the cancellations took hold. This shift in attitude highlights the importance of community interaction and the role of rituals in fostering social connections as well as the connection with the sacred (sometimes the community itself). In the words of a Christian Science student: "It is also important to be present. When we left the service we had a coffee, so the affection, the love, the hugging, all of that is also important". A leader of the Brethren Assemblies highlights the meaning of "church" as "People who come together. If it is not like that, the church does not exist. For the church, gathering together is vital, not only important". This aspect is key for those who have a sacramental sense of community, as a member of the Pentecostal and Charismatic Fraternity put it: "The hardest thing was,

in the communion, the fact of not being able to enjoy ourselves, because an embrace does not cease to be healing". A Christian Orthodox priest also complained: "Communion is very important because the body and blood of Christ make the union of the Church. The faithful gather around Christ, and for Christ. It is very important for the faithful to receive communion. People were deprived of communion in this time". Finally, some communities also stressed the unique efficacy of their rituals as they are. Embedding this idea in her argument against the online replication of the prayer, a member of the Islamic community told us that:

> The issue is that it [prayer] is not valid if it is not physical. We made a consultation. In the Muslim religion you must have references to the behavior of the prophet and the Koran, but there are no efforts of the wise men on these issues and they said "no, by social networks they are not valid". Then, on Fridays, as prayer are shared in a group, it was said "each one should pray at home, make the sermon with your children, but it is not valid to pray with the radio". Wise men said: "What would happen if the network is cut? It is better to do it alone at home, with the family.

The domestic confinement was considered both a positive and negative experience. Different members of the Sunni community remember their domestic Ramadan in very different fashions. Two different young women described their experience in positive terms. One said: "It was the first Ramadan we spent as a family, praying with our children. It is not only my experience, my friends felt the same"; and the second one said: "We were all able to be at home, eat at home, pray at home, we could not do that in other Ramadans". In contrast, a young man who lives alone in Madrid, told us that Ramadan at home was terribly sad for him:

> A lot of people preferred eating at the mosque to eating at home. Me, for example. My family is in Egypt and you feel sad when you eat, you don't feel like it, and that's very close when a lot of people are eating with you, you can tell when you are alone or when you are with a lot of people. Imagine the sadness when we were cooped up at home. [In the mosque] you try many foods (Egypt, Palestine, Morocco). And usually there is also a day that falls in the last ten days of Ramadan, when people come to the mosque on the 27th, people even pray outside, we make mosque open all night, they pray and have food to start the day. We have a party; we give out presents . . . Now we can't do it, at home it has another meaning, with COVID-19 you feel a tremendous sadness.

We found similar considerations and contrasts in other people and communities and, for many, the memory of the domestic leads to the memory of the virtualization. Virtualization played an important role in keeping the rates of participation and the size of the community stable. Those communities that refused the virtualization (mainly Muslims, Orthodox and some Adventists) referred a severe decrease in their regular members. An imam told us that: "There are people who no longer go to mosques. There are fewer people than usually, and . . . we have not had to throw anyone out: we have fought for them to come!" An Adventist leader conjectured that: "The members of the churches sought out virtual worship services, and if their churches did not offer them, they sought out others who offer them". This idea is partially confirmed by the fact that all the religious communities with regular virtual activity found an increase in their numbers on their online services in the period of the most severe restrictions. A pastor of the Philadelphia Church said that "the number of people accepting Christ has multiplied. We have a Zello in Madrid that has a 24-h broadcast, and there have been up to 500 people in one night who have accepted Christ as their savior." A representative of the Spanish Reformed Episcopal Church describes his perspective of successful virtualization like this:

> The digital shift meant an exponential growth, it has been extraordinary. Since the early days, when the congregations were broadcasting with a cell phone, until few months later, when ( . . . ) we had a radio station that has gone from 10,000 to

110,000 listeners in the course of the epidemic, transmitting the services in five languages ( … ); our YouTube channel has 180 hymns recorded ( … ). These resources were uploaded to the networks, and they are extraordinarily positive. Our YouTube channel and the web page have collected thousands of visits, and the official religious services of the denomination are re-broadcast for those places where there are believers and the churches are closed.

Despite the numbers, some leaders considered the experience of preaching through the Internet as lonely. A Baptist pastor expressed how he missed the closeness of in-person counseling saying that: "The most suffering work has been pastoral work, not being able to visit the homes, the personal relationship in the pastoral office, having that coffee with the brother who needs help, who needs guidance". Similarly, another pastor of a small evangelical community told us that:

I personally felt very bad talking to the void, preaching to empty chairs … I felt a deep sadness. That's a personal thing for me. There was a deep sadness in seeing me myself preaching to something dead, a dead space, the same space where we were congregating, singing to the Lord. I happen to be speaking to a wall.

After the reopening of worship centers, prophylactic measures became even more crucial than before. By that point, the majority of the population had become used to wearing masks, practicing social distancing and using sanitizers, and these measures were widely accepted as effective means of restoring community life. In our research, we found that there were no religious groups in Spain that explicitly opposed anti-COVID measures; although, certain members of various groups did express skepticism about the pandemic. Despite this, religious leaders and organizations did not engage in public discussions about these measures. While the Evangelical Federation fought against the prohibition of chanting (which was only approved in two regions), this same institution and many others supported and promoted prophylactic measures actively. However, for churches and groups with a sacramental sense of ritual, social distancing and hygiene norms still posed a challenge. An Episcopalian Bishop told us:

Spiritual communion you can maintain through the media … But communion is more than spiritual: it is physical and it requires physical actions, being together with people, seeing them fully. [ … ] Now, in part, it has been alleviated when they can go back to the churches. But how can they return to the churches? Well, they can't partake of the chalice … And this thing of touching each other, having a conversation, going to the end of the service and accompanying a person to the corner, having a coffee with people after the service to continue to maintain that communion. It's been heartbreaking.

Finally, if cancelations were considered generally negative, the outdooring of activities was generally well received. Everyone we interviewed agreed that they welcomed the access to the streets, parks and open spaces as a relief after many weeks of not even being allowed to go for a walk. However, this opening was very slow and with many regulations as well. Before long, the new regulations on the use of open space again proved to be burdensome and a source of other problems. For example, for many months it was forbidden for groups of more than six people to gather in both open and private spaces, which affected all kinds of family and community celebrations, such as Christmas. Curfews were also maintained, especially affecting collective Ramadan prayers. Some groups that gathered in public spaces were afraid of receiving anti-cult discriminatory reactions for their practices, and stopped doing so. Parks and natural spaces were so crowded during these months that pagan and African-matrix groups who wanted to access their usual spaces for nature rituals were unable to do so due to a lack of privacy and appropriate places.

## 3. Elasticity and Ritual

According to Grace Q. Zhang (2015), elasticity is the "springy" property that allows the stretching of linguistic terms and expressions in multiple manners and contexts to satisfy

emerging discursive needs. In a metaphoric sense, stretching means "adjust, modify, and manipulate our words" to accommodate a satisfying communicative interaction (Zhang 2015, p. 5). In linguistic communication, elasticity is strategic, as it seems to be in rituals, which means that it must have a goal. For Zhang's model, the goal is, typically, communication, but for the purpose of a ritual analysis, we cannot assume that communication with the sacred or with the religious community is always a goal; although, it might be. According to our definition of ritual, the proper goal of ritual behavior is to organize the experience of the sacred, whatever each community might define as sacred and order, for that matter. Thus, the adaptations we found in 20 denominations in Spain seem to have been ruled by the general purpose of ensuring a proper experience of the sacred, at the same time that the measures against COVID-19 forced the communities to stretch, adjust and negotiate the ways of organizing that experience.

### 3.1. Maxims for the Ritual Change

The variety of the adaptations, and the subjective experience of them, reveals that the main goal of ritual elasticity went with some specific goals that each community considered according to their own perspective of their practice. In 2011, Zhang identified four different specific goals of this elasticity that she identified as "maxims" and they are useful for our case too: "(1) Go just-right: provide the right amount of information ( . . . ); (2) Go general: speak in general terms ( . . . ); (3) Go hypothetical: speak in hypothetical terms ( . . . ); (4) Go subjective: speak in subjective terms" (Zhang 2015, p. 579). These maxims constitute a dynamic set of orientations, operating simultaneously (although in different degrees) when we are choosing the terms and expressions through which we reach a successful communication. In 2015, the author renamed and reorganized the maxims in a different scheme, but for the purpose of the ritual analysis, we find the previous version more accurate.

Jumping from linguistics to ritual analysis, we can see the maxim "go just-right" whenever the religious communities discuss the proper way of implementing the experience of the sacred. Ideally, this is the goal that put formal efficacy in the first place but, again, different groups identified what was just right for them: for many Christian groups, the online replication of the communal worship was an excellent way of keeping their preaching at the center of the community life; for others, such as Sunni Muslims, private prayer was more efficient than online prayer in order to be closer to the sacred; and other groups reinforced their sense of service by devoting themselves to charity work while the worship places were closed. However, for others, there was no way of going just-right without communal prayer, or communion, or a proper gathering (for weddings), or particular objects (water), which lead them to choose cancelation or rescheduling. In other cases, such as funerals, when cancelation was not possible, the going just-right was truncated because of the severe space limitations, and the feeling of rituals going wrong spread.

"Go general" seems to have been an important orientation for those groups that increased their numbers through the virtualization of their worship and other services, as well as for those who chose to keep the community together (through online gatherings) over the rightfulness of the rites. The other two maxims just left the door of the ritual open to improvisation. For the first weeks of the lockdown, both cancelations and adaptations seem to have gone hypothetical just to try different solutions to the closure of the worship centers. Going subjective in a ritual context might be understood as some individual freedom to celebrate and practice, which is compatible with forced confinement to domestic spaces. However, the groups that more enthusiastically embraced the going subjective were the same that were already open to improvisation and individualization in rituals, such as Pagans and followers of African-matrix traditions.

### 3.2. Stretching Factors

Zhang's theory also offers an interesting perspective about the elements that articulate the changes and torsions when elasticity shows up. If language does have vague words and



expressions that make elasticity possible, can we find ritual elements or dimensions that play a similar role? In ritual theory, vagueness has been considered as ambiguity (Flanagan 1985; Engelke 2006), and even if some traditions are devoted to formal rigidity, it would not be rare that most of the rituals present some undefined element or dimension that are eventually open to change. In 2015, Zhang identifies at least 14 different "stretchers" that apply to discursive communication. For our case, since stretchers will bring some sense of vagueness, the challenge of this approach in ritual studies consists of paying attention to what is not fixed, what is flexible, and even volatile, in a particular tradition. Ultimately, stretchers point out what is changeable, or even disposable. This implies going against the traditional perspective that thinks of rituals as fixed and rigid, but we hope it can provide a better idea of how rituals change. The broad sample of communities studied suggests that there are at least three elements that bring ambiguity and articulate ritual elasticity:

Quality stretchers deliver the experience of the sacred but, at the same time, are not essential to the experience. Pastors in many Christian denominations are important participants in the ritual delivery of the experience of the sacred, but they are not essentially linked to them as priests are in Catholic or Orthodox denominations. Counselling is an important vehicle of the experience of the sacred for many communities, but it is key for Scientologists. A particular place of worship is not usually essential for the religious experience in the largest communities, but it is essential for pagans and other religions focused on nature. The same line of consideration can apply to objects, times, actions, etc.

Quantity stretchers that adjust the frequency of the practice and the size of the participants are also in play for our case. Five prayers per day are mandatory for Muslims but beyond that, how many times they do things other than the prayer (profession of faith, charity, etc.) is flexible. One mass in a week is mandatory for Christians, but how many times they pray is up to them, usually. Vaishnavas Hindus chant 16 rounds of Mahamantra every day, but there is not a fixed number of meals, which are sacred too. How many participants in the experience of the sacred might have the same approach, as well as all other quantifiable elements?

Satisfaction stretchers enlarge the ritual experience of the sacred according to a subjective feeling of satisfaction/dissatisfaction, orienting what feels good (or not) to change. Closing worship places felt generally ok, as well as the outdooring some months later, even if they did not facilitate contact with the sacred. Some Christian churches felt happily satisfied with the virtualization of the service and how it increased the number and the frequency of contact with the sacred, others felt online services were "dead space". Some Muslims enjoyed immensely Ramadan at home, others felt terribly sad. Some felt prophylactic measures were safe and healthy for the community, others complained that chalices could not be shared as it is customary.

Zhang's theory is rather complex and offers many other elements of interest to explore from the perspective of ritual and religious change, improvisation, creativity and innovation. This includes her insights about vagueness and its roles, or the interconnection between maxims, stretchers and pragmatic functions, for example. At least for now, it seems that her theory helps to address some elements (such as maxims and stretchers) that are somehow transversal to the considerably broad range of ritual diversity. That is, on its own, noticeable, and according to Zhang's formulation, there might be universal factors of ritual elasticity, even when they are also embedded in contexts and cultures. Of course, further research and analyses are needed in order to explore this.

## 4. Conclusions

Religious change is much more than secularization surveys and religious switching; it includes the variation of those elements and actions that so often we have considered fixed and rigid, the transformation of rites. In order to explore this transformation, we applied a linguistic approach, which is a classic strategy since the debt of ritual studies with linguistics and semiotics is as old as it is fruitful. Our purpose was to contribute to the contemporary literature on creativity and innovation that is changing the classic

approach and notion of ritual as fixed, rigid, traditional, etc. In doing so, we also tried to develop a wide multireligious perspective that is not usual for these matters. As a result, we state that the role of religion in challenging times cannot be reduced to the mere functions of coping mechanisms (the theodiceal theme) or making sense of the irrational (secular theme). Challenging times give us the opportunity to explore religion and ritual with a new perspective that pays attention to how innovations happen, what are the goals that lead them, what features they bring, etc. While more research is needed, our analysis of ritual adaptations to COVID-19 measures in Spain suggests that there might be universal factors of ritual elasticity, even when they are also embedded in contexts and cultures.

**Author Contributions:** M.C.-V. and B.M.-A. contributed equally to this paper. All authors have read and agreed to the published version of the manuscript.

**Funding:** This research was funded by the Foundation for Religious Pluralism and Coexistence, grant reference COVMINREL/2021.

**Institutional Review Board Statement:** The study was conducted in accordance with the Declaration of Helsinki, and approved by the Institutional Review Board GINADYC of Complutense University of Madrid (Ref. GINADYCUCM022021, 2 June 2021).

**Informed Consent Statement:** Informed consent was obtained from all subjects involved in the study.

**Data Availability Statement:** Data are unavailable due to privacy and ethical restrictions.

**Conflicts of Interest:** The authors declare no conflict of interest.

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
