# Peer review of "Elastic Rituals: A Multi-Religious Analysis of Adaptations to the COVID-19 Crisis"

_religions, doi:10.3390/rel14060773_

Round 1
Reviewer 1 Report
The article is entitled to be published as it has relevance to the issue of covid and religion in the European context. However, I miss the elements, descriptions, and insights I keep listing below.
First, in the fragment "to the point that 'elastic' and 'ritual' might seem to be antithetical terms", when the referred subject should be developed more, and what would be "antithetical terms?" It's not been cleared.
Second, in the text development, the author writes “although it is more generally devoted to the discussion about the role of religiosity in the population response to health policies, social distancing, prevention measures, and vaccination”, when this subject was extremely delicate in countries like Brazil, albeit in part of Europe, when religion, governments and vaccination were stressed, as it was written in Fábio Py, “Bolsonaro's Brazilian Christofascism during the Easter period plagued by Covid-19”. International Journal of Latin American Religions, v.4, 2020: https://link.springer.com/article/10.1007/s41603-021-00143-5.
Third, in the part “determined context and period of history, the multi-religious perspective remains under”, it is worth going into more detail about what would be an underestimated multi-religious perspective. It is not clear what you are referring to.
Fourth, it is also worth being clearer on the sentence “According to the data we gathered, rituals are not communicative per se, but they are part of complex systems of practices and relationships, sometimes communicative, sometimes not”.
Fifth, on the matter of the plurality of practices in national territories with Covid-19 and religion, as written in the sentence “There was never a law that required the closure of places of worship, but the vast majority of them did close their doors in order to facilitate the containment of the virus through in-person contact”. I suggest incorporating the article by Frank Suaski & Fábio Py, Religion and the Pandemic—Latin American Responses, International Journal of Latin American Religions 4 (2020): 165-169; where they deal with the different approaches to Covid-19 in Latin America when the authors highlight the difficulty in standardizing treatments for the pandemic within national territories. However, in each way, all territories adapted to the pandemic context, outlining possible solutions during the health crisis, and promoting changes in their rituals for the very same reason.
I do believe that the article deserves to be published as long as it considers these five changes to provide more density to the material or to clarify the article’s scope. Thus, the article deserves to be published as long as it assumes the elements indicated in the opinion.
Author Response
Thank you for your kind and insightful revision.
About the first point, we added some clarification to express how "ritual" and "elastic" might seem antithetical categories, referring to the tradition in ritual studies that considers ritual as the opposite of flexible or elastic.
About the second point, we found the recommended text extremely interesting since some of our work (not referred here) is related to the christian far right rising. We appreciate the reviewer made us aware of this fascinating analysis that we can soon add to another paper we are working on. Despite the undoubted interest of the article, we believe that it does not fit well with the analysis of the ritual adaptations we make here, as they are somewhat distant topics, only secondarily connected. We found that the other text recommended by the reviewer, on the contrary, is an excellent background for our research and so we added to the text and our references.
About the third point, we added a short explanation about the scope of the literature focused on only one and more than one denominations, in order to show how the multi-religious perspective is unusual (in fact, there is a clear lack of it).
About the fourth point, we added an extra paragraph to explain how and why we do not consider rituals as communicative acts, but behavioral devices, with some concrete examples to illustrate the perspective.
About the fifth point, we again appreciate the recommendation of this text by the reviewer. The text is an excellent addition to our reference list and we also mention it in the text, in the introduction section.
Thanks again for this comments and recommendations that we believe they improve our text and wide our research significantly.
Reviewer 2 Report
it is a good contribution
Author Response
Thank you very much for your kind review!
Reviewer 3 Report
This is an excellent manuscript, which proposes an analysis of the changes in the exercise and characteristics of religious rituals to which minority religions have been forced in Spain since the COVID-19 lockdowns.
Its analytical proposal stands out, recovering concepts and theories from Linguistics to demonstrate the elasticity and fluidity of rituals.
The author(s) should be more explicit in to describing the sample selection criteria, and how they developed the interviews.
At the same time, I've noted that Zhang's texts are not included in the References section. They should be added.
But, overall, great work!
Author Response
Thank you very much for your kind review!!
First, the reviewer asked the authors to be more explicit with the sample criteria and the interviews, and so we added a new paragraph explaining these two questions in the first paragraph they are mentioned.
Then, the reviewer makes us aware that the Zhang's references are missed in the list of reference, and so we added the two references missed, Zhang 2011 and 2015.
Thanks for your comments!!
Reviewer 4 Report
As presented, this article is flawless. According to the literature of the 90’s, “flexibility in rites is not a characteristic of contemporary religions.” However, some recent literature found flexibility. The purpose of this article is “to contribute to the contemporary literature on creativity and innovation” by providing examples of adaptation during the Covid-19 crisis. Adaptation is further explained in terms of verbal flexibility. This article is well documented and provides ample empirical evidence.
For those who expect more than just empirical data, this article is void of theoretical reflection. There is no distinction between “facts” and their interpretation..
For Grimes, the “conception” or “interpretation” of anthropologists studying traditional rigid societies is that rituals are rigid – but anthropologists present separately their data and their interpretation. By contrast while 40 interviewees present their interpretation of rituals as flexible, the author writes as if the rituals themselves were flexible (as in table 1).
There is also no distinction between language and reality, a major issue of the philosophy of language and rhetoric. All human language is “elastic” but traditional rhetoric recognizes various degrees of analogy, some being unacceptable. Analogically speaking, one may that a mouse is a small size elephant; this is an unacceptable analogy unless one can show what mice and elephants have in common. The author provides numerous examples of different adaptations, but he/she does not show what the old and new have in common, simply saying that there is language “stretching.” So, what is being stretched, is it the language or the rituals?
“Stretching means ‘adjust, modify, and manipulate our words’ to accommodate a satisfying communicative interaction.” This is a national issue today in the U.S. To accommodate half the American population, objective reporting is called fake news, lies are an accommodated form of truth, and conspiracy theories are presented not as theories but as facts. Zhang provides various “maxims” (e.g., “speak in subjective terms.” In subjective terms, my conspiracy theories are facts and your facts are lies to me). This article'sdiscussion takes place in a philosophical and rhetorical void. By stretching or not stretching language, rituals themselves are seen as either flexible or rigid.
Yet, as standards of empirical research go, this article passes the bar.
Author Response
Thank you very mucho for your kind and insightful comments!!
The reviewer does not recommend any change or revision, but a very interesting reflection is added, pointing at the controversial matter of fake news and the contemporary crisis of truth and credibility that is directly related with how we, scientist, provide a proper differentiation between facts and interpretations. That is certainly a very important issue raised by postmodernist, whose consequences are now uglier than expected before. We appreciate greatly this comment and we will consider this topic for some analysis in the future, since we are already involved in a project about science and spirituality that aims precisely to that kind of questions.
Round 2
Reviewer 1 Report
Tranks for submitting and I’m glad I helped with the comments.
The paper has been duly augmented with the additions.
Only one detail is correct quote is:
(Usarski and Py 2020)
Usarski, F, Py, Fábio. 2020. Religion and the pandemic – Latin American responses. International Journal of Latin American Religions 4 (2): 165-1969.
And not:
(Suaski & Py 2020)
Suaski, Frank and Fábio Py. 2020. Religion and the Pandemic: Latin American Responses. International Journal of Latin American Religions, 4(2): 165-169.
Tranks.